# Spina Bifida: A Review of the Genetics, Pathophysiology and Emerging Cellular Therapies

**DOI:** 10.3390/jdb10020022

**Published:** 2022-06-06

**Authors:** Abd-Elrahman Said Hassan, Yimeng Lina Du, Su Yeon Lee, Aijun Wang, Diana Lee Farmer

**Affiliations:** 1Division of Pediatric General, Thoracic and Fetal Surgery, University of California Medical Center, Sacramento, CA 95817, USA; suyle@ucdavis.edu (S.Y.L.); aawang@ucdavis.edu (A.W.); 2Center for Surgical Bioengineering, Department of Surgery, University of California Davis, Sacramento, CA 95817, USA; lyidu@ucdavis.edu

**Keywords:** pediatric surgery, fetal surgery, stem cells

## Abstract

Spina bifida is the most common congenital defect of the central nervous system which can portend lifelong disability to those afflicted. While the complete underpinnings of this disease are yet to be fully understood, there have been great advances in the genetic and molecular underpinnings of this disease. Moreover, the treatment for spina bifida has made great advancements, from surgical closure of the defect after birth to the now state-of-the-art intrauterine repair. This review will touch upon the genetics, embryology, and pathophysiology and conclude with a discussion on current therapy, as well as the first FDA-approved clinical trial utilizing stem cells as treatment for spina bifida.

## 1. Introduction

Spina bifida (SB) is the most common congenital defect of the central nervous system. While survivable, the defect portends lifelong disability to all those afflicted. Myelomeningocele (MMC) is the most severe form of SB and is characterized by an extruded spinal cord contained within a sack of cerebrospinal fluid. Closed neural tube defects (NTDs), a much less severe defect, exist and are often associated with lower morbidity than spina bifida itself [1]. Recent studies have demonstrated an incidence rate of approximately 3.63 per 10,000 live births in the United States and an incidence rate of 18.6 in 10,000 worldwide [2,3]. Furthermore, while the mortality rate has declined to approximately 8% for liveborn infants with SB, this is still over 10 times higher than the national average of all U.S. births [4,5]. Those who survive are afflicted with motor impairment, bowel, and bladder dysfunction, as well as neurological sequelae. In particular, Arnold-Chiari II malformations (the downward displacement of the cerebellar vermis and tonsils) are associated with SB and are known to have effects on motor, cranial nerve, and cognitive functions—all of which impact the quality of life for SB patients and their families. 

Our understanding of NTDs has improved dramatically over the last several decades, but questions on mechanism of formation, preventative measures, and curative treatments remain. An intricate relationship between environmental, genetic, and maternal factors has been implicated in the etiology and pathogenesis of this congenital malformation [6,7]. Currently, the genetic component is thought to contribute to approximately 60–70% of the risk of NTDs and, as such, the identification of additional genetic loci in humans is an ongoing area of study [8]. This is underscored by the approximately 250 genes identified in the murine model of NTDs alone, ranging from spontaneous, chemically induced, and knockout variations [6].

In this review, we will discuss the genetics and embryology involved in SB and NTDs, explore folate metabolism, briefly touch on the pathophysiology and surgical repair options, and review the previous and current work in stem-cell-based therapies. 

## 2. Embryology

The neural tube is a transient structure that is formed during the development of an embryo; it is the precursor to the central nervous system, which is composed of the brain and spinal cord. In human embryos, this entire process occurs between days 17 and 28 after fertilization [7]. The activities required to form a normal neural tube include apoptosis, neural crest migration, neuroepithelial proliferation, contraction of apical cytoskeletal microfilaments, and flexing at dorsolateral bending points [8]. Any aberration during embryogenesis can result in neural tube failure and a resultant NTD. Neural tube closure can be succinctly summarized by sequential folding, elevation, closing, and fusing of the neural tube along the dorsal midline, allowing for functional separation of non-neuronal tissue from the neural tube itself. This entire process and sequence of events is termed neurulation, which can be further subdivided into primary and secondary neurulation [9].

During primary neurulation, the brain and spinal cord are formed, specifically the segments extending from the medulla to the mid-lumbar enlargement. Primary neurulation is responsible for the shaping, folding, and fusing of the neural plate along the midline [9]. In mouse embryos, the act of primary neurulation is initiated at the boundary between the cervical spine and future hindbrain. Closure spreads bi-directionally from this point. A second closure origin site arises at the boundary of the forebrain and midbrain; this closure also spreads bi-directionally. A third closure event originates at the rostral end of the forebrain as well. Closure between all three of these initiation sites leads to completion of cranial and spinal neurulation, thus providing a closed anterior, posterior, and hindbrain neuropore [9]. These events, as seen in mice, are suggested to be similar in human embryos with some key differences [10,11]. Notably, the existence of a second closure event (Closure 2) remains uncertain and brain formation may occur as a direct progression of neurulation from Closure 1 to Closure 3 initiation sites [12]. This may be due to differences in the position of Closure 2 in human populations, as seen between mouse strains, or that primates have evolved to not require a Closure 2 initiation site given a smaller brain at the corresponding stage as compared to mice [12,13].

Aberrations in neural tube closure during primary neurulation, can lead to a host of NTDs, including SB. The discontinuous nature of neurulation leads to specific defects, depending on where the faults occur and on how much of the neural tube remains unfused. As noted, the summation of all three closure sites results in a complete cranial and spinal neurulation and, therefore, a closed anterior, posterior, and hindbrain neuropore. Faults in the initial closure site lead to defects along the length of the midbrain to the lower spine, creating a condition known as craniorachischisis (absence of the brain and cranial vault with a contiguous bony defect of the spine). Anencephaly (absence of major portions of the brain, cranium, and scalp) results from failure or abnormalities in the cranial neurulation process. Spina bifida is a result of incomplete caudal neurulation [1].

Secondary neurulation is responsible for the lower portion of the spinal cord, which includes the distal lumbar cord, conus medullaris, and filum terminale. This process is composed of canalization and retrogressive differentiation. Canalization is the process by which the neural tube elongates caudally toward the posterior neuropore. As the notochord and neural epithelium fuse, a caudal cell mass is formed. Within this cell mass, multiple microcysts coalesce and form an ependyma-lined tubular structure, which then fuse with the neural tube from above, creating a continuous primary neural tube [7,14]. In humans, the development and closure of the neural tube is complete 28 days after conception. After day 38, there is controlled cell necrosis along the caudal neural tube, which results in the formation of distal lumbar cord, conus medullaris, and filum terminale. This process of retrogressive differentiation is responsible for removing much of the caudal cell mass; unlike humans, in mice, the tail bud develops for eventual formation of a tail [14]. Disorders in secondary neurulation can also lead to other congenital lesions, including caudal agenesis, presacral cysts, hindgut abnormalities, caudal lipomas, and a myriad of anorectal and genitourinary anomalies [15,16]. A more recently named lesion, the retained medullary cord (RMC), described as an elongated spinal-cord-like structure that continues to the cul-de-sac, has been attributed to either a complete or partial arrest of secondary neurulation [15,17]. It appears as a low-lying conus, but with a nonfunctioning component. While originally thought to be rare, they are more often encountered in the clinical setting and can be associated with the previously mentioned congenital lesions. Patients with disorders of secondary neurulation typically present with incontinence, frequent urinary tract infections, neurogenic bladder, and urodynamic abnormalities, as well as neurologic deficits [16].

There exist two proposed mechanisms for NTDs. First, the neural tube fails to close due to abnormalities in cellular behavior (inefficient proliferation, disorganized cellular death, and poor collective cell movement). Second, the closed neural tube reopens, possibly due to a breakdown in critical cell–cell adhesion junctions [18,19].

## 3. Genetics

Despite family history being a risk factor for SB development, recurrence patterns are not attributed to a single genetic locus. Instead, SB is a complex trait caused by a combination of variants at multiple loci and involving multiple genes [20].

### 3.1. Neural Tube Defects in Animal Models

Over 240 mutants and strains have been identified for NTDs in mice [6]. Of the mice NTDs, some cellular processes have been identified, including planer cell polarity (PCP) pathway, cell apoptosis, DNA transcription, and the canonical Wnt/beta-catenin pathway.

Apical-basal polarization of cells is critical for the morphogenesis and function of mature tissues. *VANGL1* and *VANGL2* are mammalian homologs of a Drosophila gene required for PCP to develop eye, wing, and leg tissues [21,22]. The genes *VANGL1* and *VANGl2* are expressed at the dorsal portions of the neural tube and are essential for vertebrate morphogenesis [23]. Song et al. demonstrated that PCP is crucial in linking the anterior–posterior patterning information to the left–right asymmetry of the notochord prior to leftward nodal flow across the posterior notochord [23]. Therefore, notochord cells lacking the expression of *VANGL1* and *VANG2* have random cilia distribution, leading to turbulent nodal flow and disrupted symmetry [24]. Torban et al. also demonstrated that only mice heterozygous for *VANGL1* and *VANGL2* mutations showed profound developmental defects, including severe craniorachischisis [25].

The establishment of PCP involves the Wnt/Frizzled-PCP pathway. In the drosophila model, the proteins include cadherin Flamingo, transmembrane protein Frizzled (Fz), and cytoplasmic proteins Dishevelled (Dsh), Diego (Dgo), and Prickle (Pk). Mutations in these proteins lead to loss of polarity and cytoskeletal rearrangement [26]. Furthermore, the mouse homologs of Flamingo (CELSR), Dishevelled (DVL), Frizzled (FZD), and Prickle (Pk) have all been shown to contribute to the etiology of NTDs [27]. Non-core PCP tissue specific effectors also play an important role in the development of NTDs. The Fuzzy planar cell polarity (FUZ) gene, a tissue-specific effector, is involved in directional cell movement and is essential for neural tube formation in mice and humans [24]. A study demonstrated that Fuzzy knockout mice exhibited NTDs and proposed that mutations in the FUZ gene may account for human NTDs. The mechanisms of PCP mutations in NTDs mentioned above are further studied by Humphries et al.; the group introduced the mammalian mutation into Drosophila and revealed different defective phenotypic and functional behaviors, suggesting a causative relationship between PCP and the NTDs [28].

Other mutations leading to SB in mice involve DNA transcription and apoptosis. Specifically, the Foxc2 (MFH-1) transcription factor is suggested to mediate the extensive skeletogenesis in cells derived from neural crest and the mesoderm [29]. The Traf4 gene is involved in apoptosis. Mutations in this gene lead to the inability of nucleosome assembly proteins to bind to condensing chromatin in apoptosis and dysregulation of neuronal cell proliferation [30].

Lastly, genetic mutations in the canonical Wnt/beta-catenin pathway have been studied in the SB mouse model and shown to be implicated in human SB. Regulators for bone formation during skeletal growth and remodeling include the lipoprotein-receptor-related protein 6 (LRP6) in the canonical Wnt/beta-catenin pathway. It is implicated in the growth of axial skeleton and neural tube structures [31]. Lei et al. have found that several single nucleotide variants in LRP6 predispose embryos to NTDs [32]. The mediator complex (MED) is also known to have significant regulatory effects on WNT signaling and is associated with neural tube closures [33]. Recent studies using CRISPR/Cas9 technology to generate knock-in mice models provided strong evidence between functional variants of MED genes and some NTD etiologies [34]. The canonical Wnt signaling is also implicated in the regulation of PAX3 expression and as a target of PAX3. Mutation of the PAX3 transcription factor leads to defects in neural tube closure and is preventable by folic acid [35]. Recent studies show beta catenin gain of function and PAX3 loss of function produces additive effects on NTDs due to their interactions [36].

### 3.2. Human Genetics

Depending on the affected gene, SB inheritance patterns vary in humans. Studies have indicated autosomal dominant inheritance of SB and other NTDs at multiple loci, including the *VANGL1* gene, *VANGL2* gene, *FUZ* gene, the *CELSR1* gene, and the *TBXT* gene [32,37,38,39]. Despite the identification of the single genes, it is important to note that more and more recent evidence points toward an omnigenic model of spina bifida, suggesting that human spina bifida is caused by a series of genetic variants and their interaction with environmental factors [40,41].

Various heterozygous missense mutations of the *VANGL1* gene have been identified and associated with various NTDs in humans [39]. In a review by Marello et al., a strong association between the rare variants of *VANGL1* and the NTDs was suggested. In addition, Lei et al. and Kibar et al. sequenced the *VANGL2* gene in populations with various forms of NTDs and suggested that the mutations in *VANGL2* gene may predispose human fetuses to NTDs [32,37]. Other studies proposed indirect mutations affecting VANGL2 function associated with NTDs. The heterozygous frameshift mutation of the *CELSR1* gene is found to be associated with SB in a study of 192 patients in California [32]. Its mechanism is based on the interaction between the CELSR1 protein and the *VANGL2* gene, since the mutation results in less recruitment of VANGL2 for cell-to-cell contact. Other heterozygous missense mutations in the *FUZ* gene have been identified and linked to NTDs in the Italian population; the research team found five missense mutations in the *FUZ* gene that interfere with cilia generation, cell directional movement, or both [42].

Other mutant genes that are risk factors for spina bifida include the *CCL2* gene. The *CCL2* gene controls the monocyte chemotactic protein-1 (MCP) export levels after treatment with interleukin-1-beta in vitro [43]. Research studies have shown that first-trimester hyperthermia is associated with a twofold increase in spina bifida [44]. Thus, inflammation and elevated body temperature, especially signaling by molecules such as chemokines, may be involved in the causation of spina bifida. Jensen et al. found that polymorphism of the CCL2 A(-2518)G promoter is associated with spina bifida and it is speculated that this is due to the alleles’ less intense systemic and local response to infection [45].

Furthermore, a *TBXT* mutant gene, specifically the transmission allelic variant TIVS7-2 [38], has also been linked to meningomyelocele. TBXT is encoded by the T gene and is vital for the formation and differentiation of posterior mesoderm and axial development in vertebrates. Although further studies showed that the mutant TIVS7-2 is associated with an increased risk of SB, the pathogenic mechanism remains unclear [46,47].

Since folate supplementation has been shown to prevent up to 70% of NTDs in humans, the gene mutations of enzymes involved in homocysteine-folate metabolism have also been studied extensively in humans [48,49]. These metabolic enzymes include methylenetetrahydrofolate reductase (MTHFR), methionine synthase (MS), cobalamin coenzyme synthesis, and cystathionine b-synthase (CBS). The SNP R653Q polymorphism in the MTHFR1 gene has been observed in NTDs in Irish and Italian populations [50,51,52]. Other studies have shown that MTHFR 1298 A-C combined with MTHFR 677 C-T alternation increases the risk of spina bifida [53]. Similar effects have been observed in the polymorphism of the MS and 5-Methyltetrahydrofolate-Homocysteine Methyltransferase Reductase (MTRR). Doolin et al. concluded that variants in MS 2756A-G polymorphism and MTRR 66A-G polymorphism increase the risk of spina bifida by maternal genotype [54]. Genetic–nutrient interaction has also been considered. Christensen et al. found that MTHFR polymorphism and low folate status combined is associated with a greater risk for NTDs than either variable alone [55]. It was also shown that the 66A-G polymorphism in the MTRR gene combined with low levels of serum B12 in mothers and children increases the risk of spina bifida [56].

Next-generation whole-exome sequencing (NGS) allowed the identification of variants in new candidate genes that were previously not implicated in SB in humans. Through NGS, loss-of-function de novo variants of *SHROOM3*, *PAX3*, *GRHL3*, and *MYO1E* genes are linked to the development of NTDs; their proposed mechanisms include generation of protein-truncating variant and transcription factor defects [57,58,59]. Recently, rare and novel de novo variants in *RXR**γ*, *DTX1*, and *COL15A1* genes and X-linked recessive variants *ARHGAP36*, *TKTL1*, *AMOT*, *GPR50*, and *NKRF* were also found to contribute to NTDs [60]. Furthermore, with exome sequencing, more evidence links the genes of the PCP pathway to the development of SB and craniorachischisis. The identified genes include *CELSR1*, *PRICKLE1*, *FZD6*, *SCRIB*, *PTK7*, *VANGL1*, and novel genes *FREM2* and *DISP2* [61,62,63,64,65]. Due to the genetic complexities and omnigenic nature of NTDs, clinically significant genetic targets will vary from individual to individual. Therefore, genome sequencing is a crucial step towards improved understanding of the genetic implications in NTDs and precision medicine for SB.

## 4. Nongenetic Factors

### 4.1. Maternal Factors

Maternal diabetes and obesity have both been implicated in an increased risk for NTDs [66,67]. The teratogenic effects of hyperglycemia and hyperinsulinemia are thought to contribute to increased cell death in the neuroepithelium of the neural plate [68]. There is a 2-to-10- and 1.5-to-3.5-fold increase in risk for mothers with diabetes and obesity, respectively; the risk appears to increase as maternal body mass index does [1]. Moreover, this risk does not only arise in offspring of type 2 diabetes (insulin-resistant), but is also present for children of mothers with type 1 diabetes (insulin-dependent); prenatal counseling and care are crucial in this population [69,70]. Additionally, inadequate maternal nutritional status has been associated with an increased risk of NTDs; deficiencies in folate, zinc, and B12 have all been implicated risk factors; however, folate deficiency has the most support [71,72,73,74]. Alcohol and caffeine use were also potential risk factors [75,76]. As noted above, maternal hyperthermia, be it from fever or environmental factors, has been hypothesized to have a teratogenic effect upon the developing embryo [44]. A systematic review and meta-analysis by Moretti et al. found a significantly increased odds ratio of 1.92 (95% CI = 1.61–2.29) for NTDs in pregnancies affected by maternal hyperthermia [77]. While the heat source varied from maternal fever to saunas or even exercise, the elevated risk remained consistent.

### 4.2. Medications

The most well-known drug responsible for NTDs, is valproate (valproic acid), an antiepileptic drug (AED). This was first reported by Robert et al., who noticed an association with women being treated for their seizure disorder with valproate and the occurrence of spina bifida in those pregnancies [78]. The mechanism is thought to be secondary to valproate’s action as an inhibitor of histone deacetylase, which disrupts the balance of protein acetylation and deacetylation, leading to neurulation failure. The mechanism for this is thought to be due to a disruption in the Wnt signaling pathway [79]. Additional studies have investigated other AEDs (carbamazepine, phenytoin, and lamotrigine) and have found that valproate continues to have the most severe risk profile for the developing fetus; the risk appears to be dose-dependent, with the recommendation to avoid valproate, if possible, otherwise to limit its dose [80,81,82,83].

## 5. Folate Metabolism

Folates are folic acid compounds that refer to a class of essential water-soluble vitamins, mainly found in fruits and green leafy vegetables [84]. Folic acid is a synthetic compound, mostly used in pharmaceuticals and dietary supplements, owing to its chemical stability. The umbrella term “folate” refers to all these compounds, be they natural or synthetic. Folic acid (pteroylglutamate) is the most oxidized form of folate, and it must undergo reduction to be biologically active. This process occurs in two steps via the same enzyme (dihydrofolate reductase), first to dihydrofolate, then its active form, tetrahydrofolate. All proliferative cells utilize folate in this manner [84]. The addition of a single carbon unit and reduction steps produces 5-methyltetrahydrofolate (5-MTHF), which is the predominant form found in plasma [8].

The exact mechanism of how folate supplementation rescues NTDs is unclear, but one hypothesis centers around its ability to recover a disturbance in cell growth in the neuroepithelium of the neural plate and neural fold [82]. Given the rapidity at which the neuroepithelium grows, there is a high rate of nucleic acid synthesis that must be met to facilitate this growth. The active form of folate, 5-MTHF, is a critical component in nucleic acid synthesis and is generated by the conversion of tetrahydrofolate to 5-MTHF via the glycine cleavage system (GCS) [83]. The GCS is a series of enzymes that are part of the most prominent glycine and serine catabolic pathways and the generation of 5-MTHF, one of the few C_1_ donors in vertebrate biosynthesis; it is also highly expressed in the neuroepithelium [83,85]. One of the primary functions of folate in metabolism is to transport single-carbon groups [86]. These exchanges of carbon groups are used for the modification or biosynthesis of a range of biosynthetic molecules and processes, including DNA synthesis and DNA and RNA modification; folate acts as either a donor or an acceptor of one-carbon units and functions as a coenzyme involved in these processes [84]. These reactions are categorized as the folate-mediated one-carbon metabolism, key reactions that occur via interconnected pathways between the cytosol, nucleus, and mitochondria [86].

In mammals, folate is taken up from the plasma (as 5-MTHF) by folate receptors into the cytosol via endocytosis. Once in the cell, further additional glutamate residues are added (polyglutamation), which prevents this form of folate from leaving the cell [8]. *FOLR1*, a gene responsible for coding a protein involved in the transport of folate into the neuroepithelial and neural crest cells, has been studied in the murine model; the protein itself is expressed primarily in the epithelial cells of the choroid plexus, kidney, ovaries, and placenta. Defects in the cranium, neural tube, and cardiac tissue in the setting of functional inactivation of *FOLR1* were noted [87,88]. However, these defects were rescued when heterozygous embryos were supplemented with folate, providing that the decreased ability to uptake 5-MTHF could be supplanted by providing additional folate [89].

Additionally, folate supplementation is thought to prevent NTDs via modulation of epigenetic processes, such as DNA methylation [90,91]. Methylation’s role is particularly important given the extensive chromatin methylation that occurs during early embryogenesis [92]. Moreover, in the murine model, previous work has demonstrated increased rates of NTDs with methylation disruption [93,94,95]. Disruptions in the methylation cycle can lead to altered levels of S-adenosylmethionine and S-adenosylhomocysteine. In the methylation cycle, S-adenosylmethionine is the key methyl group donor; S-adenosylhomocysteine is generated by a loss of a methyl group from S-adenosylmethionine [96,97]. Elevated levels of S-adenosylhomocysteine are a strong inhibitor of methyltransferases and are cleared via conversion to homocysteine [8]. If the ratio of S-adenosylmethionine and S-adenosylhomocysteine is pushed towards S-adenosylmethionine, then the cell is no longer primed for continued methylation [8]. Linden et al. demonstrated widening of the anterior neuropore in chick embryos when treated with methylation inhibitors [98]. Toriyama and colleagues demonstrated that methylation disruption also compromised normal neural tube closure and may be related to dysfunction in ciliogenesis, a key cytoskeletal process involved in neural tube closure [99]. The effects of methylation interference on neurulation and additional pathways are still under investigation.

As noted, the transfers of 1-carbon moieties result in metabolic byproducts that must be cleared, such as homocysteine. Steegers-Theunissen et al. demonstrated that a group of women with previous pregnancies affected by NTDs had elevated plasma concentrations of homocysteine, despite normal folate levels, perhaps indicating homocysteine levels as a marker for dysfunctional folate metabolism [100]. A specific mutation, C677 T, in the methylenetetrahydrofolate reductase gene is a known cause for elevated plasma homocysteine concentrations; this polymorphism has been seen to be more frequent in mothers of offspring with NTDs against controls [101]. However, the NTD incidence has not been shown to be elevated in certain murine models of hyperhomocysteinemia [102,103]. To further clarify homocysteine’s role, Yang et al. performed a systematic review and meta-analysis; they noted a slightly higher level of homocysteine in maternal plasma in mothers with offspring affected by NTDs. However, they noted the heterogeneity of blood sampling time, as well as presence or absence of folate supplementation, were confounders that must be accounted for [104]. The true relative contribution of homocysteine metabolism to NTD occurrence remains and further study is warranted.

Today, the decreased incidence of NTDs is largely due to the prevention of folate deficiency, a risk factor that was implicated as early as in the 1960s, but not fully appreciated until the late 1980s and early 1990s [105]. Numerous studies demonstrated a relationship between lower maternal folate levels and decreased incidence of NTDs; additional randomized clinical trials helped establish the 4 mg/day recommendation for high-risk mothers, defined as women with a previous pregnancy complicated by an NTD [106,107,108,109,110,111]. The methods available for increasing the consumption of folate were via alterations in dietary habits, use of supplementation, and fortification of food. In 1996, the Food and Drug Administration (FDA) authorized folate fortification and, by 1998, mandated it [112]. As of 2021, over 78 countries have mandated folic acid fortification in their flour [113]. Despite the mounting evidence of a decreased incidence of NTDs with folate supplementation, there are still many countries that forgo fortification and continue to have a significant number of births complicated by NTDs [113]. While it has been demonstrated that folate deficiency is a cause of NTDs, there exist mouse models, as well as human populations, that develop NTDs despite folate supplementation, again underscoring the complex genetic and environmental relationship involved in this process [114].

## 6. Pathophysiology

As previously noted, the primary injury in SB is due to the disorder in primary neurulation. With this process disrupted, the spinal cord, which would otherwise normally be protected, is now exposed to the fetal environment. In a series of experiments with fetal rats and pigs, Heffez et al. described the “two-hit” hypothesis of injury for the destruction and damage to the spinal cord as the prevailing theory on how the exposed cord eventually succumbs to neurological disability [115,116]. Initially, the exposed neuroepithelium will differentiate and develop properly, but the “second hits” of exposure to the amniotic fluid and the mechanical trauma of the exposed cord in utero ultimately lead to neuronal cell death and loss. The work by Stiefel et al. in mutant mouse models has supported this hypothesis, in addition to clinical observations of rapid neonatal loss of motor function, despite the presence of normal-appearing movement in utero [117,118,119]. This model for the etiology of the neurological functional defects seen in NTDs prompted attempts at closure of the lesion during fetal development in hopes of ameliorating the effects of the mechanical and amniotic trauma. Additionally, studies in a fetal lamb model suggesting that the hindbrain herniation associated with the Chiari malformation could be reversed by closure of the open spinal cord defect further supported the idea that the functional defects could be recovered by in utero repair [120].

## 7. Management

There is no single medical or surgical treatment that can completely ameliorate or recover the constellation of symptoms and deficits that develop associated with SB and MMC. However, many interventions can improve the quality of life of these patients. Initial management involves basic stabilization/resuscitation (i.e., keeping infant warm, providing oxygen, and clearing the nares if needed), with care to avoid further damage of the exposed spinal cord (if not closed prenatally) [121,122]. Attention to potential cranial nerve compromise and airway protection is paramount. The most common cause of neonatal death is airway compromise due to an inability to swallow for those with severe Chiari II malformations.

### 7.1. Chiari II and Hydrocephalus

MMC is usually associated with a Chiari II malformation, which is when a portion of the cerebellum and brain stem are pushed into the foramen magnum [114]. The malformation results from the open MMC defect that creates a traction-like effect of the brain stem from the open defect below. This results in an elongated medulla, brain stem, and fourth ventricle. The impaired cerebrospinal fluid (CSF) circulation increases pressure within the ventricles and leads to secondary hydrocephalus (build-up of fluids within the ventricles of the brain). However, hydrocephalus can occur with SB not associated with MMC or Chiari II malformation. Due to the disrupted CSF circulation, most patients will require ventriculo-peritoneal shunting (procedure to remove excess cerebrospinal fluid from the ventricles of the brain by shunting fluid into the abdomen) to alleviate the increased pressure. The need for shunting appears loosely related to the level of the lesion, with more cephalic lesions being more likely to need intervention [123]. Shunt infection and shunt malfunction necessitate many operative shunt revisions and replacements, and untreated or unrecognized shunt malfunction can lead to severe hydrocephalus and associated brain injury.

### 7.2. Bowel, Bladder, and Sexual Dysfunction

Depending on the spinal cord level(s) disrupted, many SB patients suffer from bowel and bladder dysfunction in addition to lower extremity paralysis. Urinary function has a multi-pronged approach, from using medications to improve continence, surgery to create continence, and simple clean intermittent catheterization for simple and effective bladder emptying [124]. Untreated and poorly managed bladder management can lead to hydronephrosis and may ultimately lead to renal dysfunction and the need for dialysis and kidney transplantation. As for bowel function, newborns, infants, and children will require dedicated and intense bowel management, which, in its simplest form, involves laxatives, enemas, and suppositories, but can also involve surgical intervention to provide adequate means of bowel emptying and the occasional need for colostomy.

Given the improved life expectancy of those living with SB, there is an increasing need to understand adult-related health concerns. Regarding sexual dysfunction, there are limited data on the effect of SB on sexual health and activity. Two systematic reviews by Streur et al. and Hughes et al. noted a clear trend in sexual dysfunction for men and women [125,126]. Notably, while some men (56–96%) can achieve an erection, it may not be adequate for penetration and, when present, ejaculation is often dripping or retrograde (as evidenced by semen in the urine). Only 20–67% of men report the ability to achieve orgasm as well. In women, sexual desire appears least affected, but there is a range of those with dyspareunia (pain in the genital area or pelvis during sexual intercourse), likely related to pelvic organ prolapse [127]. There is also reduced ability to orgasm. Granted, the studies that attempt to characterize sexual health in the SB patient population are limited by their small sample sizes and utilization of questionnaires not validated in this population; there is a need for engaging in sexual health dialogue at a younger age to help normalize sexual activity and sexual health [125,126,128].

### 7.3. Extremity Motor Function

Motor function impairment has been well-known to occur in patients with spina bifida; their physical impairments range from motor and sensory deficits to difficulties with stance and ambulation. The level of motor function is directly related to the level of the spinal lesion, with the ability to ambulate decreasing with more proximal lesions [129,130]. The motor deficits tend to progress with age and weight of the child. Most patients with cervical and thoracic lesions do not survive into adulthood; those with lesions at L2 and above are almost always wheelchair-dependent [129]. As for lesions in the lower lumbar (L5) and sacral level, these patients typically have better motor strength and can continue to ambulate into adulthood [131,132]. Moreover, approximately 75% of those born with spina bifida survive into adulthood. Foot deformities, such as clubbing, are also associated with spina bifida and should be addressed early by orthopedic specialists.

Overall, the complex management of patients with SB requires a multidisciplinary team involving many pediatric subspecialties (rehabilitation expert physiatrists, general surgery, neurosurgery, urology, orthopedics, and gastroenterology) and critical ancillary staff, including physical therapists, social workers, and care co-ordinators. The family’s role in managing all the physical and social aspects of this disease should also be highly emphasized.

### 7.4. Fetal Intervention

For the past century, the initial management of SB has been surgical closure within the first 48 h of birth [114,133]. The goal of postnatal surgery is closure of the open defect to minimize risk of infection. Unfortunately, in this approach, much of the pre-existing spinal cord damage, resulting from secondary injuries in utero, cannot be reversed. Early efforts of repairing these defects in the fetal patient eventually led to a randomized trial, the Management of Myelomeningocele Study (MOMS), to evaluate whether intrauterine repair of myelomeningocele between 19 and 25 weeks of gestation was superior to standard, postnatal repair [134,135,136,137]. In December of 2010, enrollment was stopped by the Data Safety and Monitoring Board after 183 of the planned 200 patients due to the efficacy of fetal surgery. The MOMS trial demonstrated a significant reduction in ventriculoperitoneal shunt placement at one year of age following fetal surgery (prenatal group: 40%; postnatal group 82%). Neuromotor function was seen to have an overall improvement, demonstrated by the finding of improvement of two or more levels of motor function leading to independent ambulation in 42% of the fetal surgery group, compared to 21% of the postnatal surgery group. A 30-month follow-up corroborated the original findings. Additionally, compared to postnatal repair, prenatal repair of MMC patients demonstrated improved scoring on a composite of mental and motor function. Secondary outcomes, including motor function and the likelihood of self-ambulation, improved [138]. The 6-year outcomes of the MOMS trials were reviewed in a secondary analysis by Houtrow et al.; they demonstrated that children with prenatal repair had better gait quality, could perform higher-level mobility skills, and were less likely to have a motor function level worse than their anatomic level [139]. Continued assessment of long-term outcomes of the patients in the MOMS trial will be crucial to determine the longevity of the results seen thus far, but also to study the potential improvement in the quality of life in these patients and their families.

Currently, there are several options for managing a pregnancy complicated by SB, including termination of pregnancy, cesarean section followed by postnatal repair, or the now state-of-the-art fetal repair for eligible patients. While fetal therapy provides significant improvement in the effects of the Chiari malformation, the improvements in distal extremity neurologic function, while promising, leave room for improvement, with 58% of afflicted children still unable to ambulate independently. Moreover, there are still many centers in the country that lack the infrastructure, training, and support staff necessary; therefore, while fetal intervention offers hope, it is still a highly specialized intervention with limited global spread.

The indications, contraindications, and techniques for fetal repair are beyond the scope of this discussion. However, a discussion of promising new treatments, specifically the inclusion of stem cells for fetal therapy, is of interest.

### 7.5. In Utero Cellular Therapies

Cellular therapy has generated significant interest in the last two decades as a potential augmentation of the prenatal repair for MMC. Initial stem cell experiments began with human embryonic stem cells, neural crest stem cells, induced pluripotent stem cells, human amniotic fluid stem cells, and, most recently, mesenchymal stem cells. Much of this work focused on murine, ovine, and avian models.

Human embryonic stem cells (hESCs) were first introduced by Lee et al. [140]. They used a chicken embryo model of MMC and injected hESCs into the amniotic cavity of a surgically created spinal open NTD both immediately and 24 h after incision [141]. The investigators found that the surgically induced NTDs were significantly more reclosed in the stem-cell-treated group compared to the control and vehicle groups. They proposed the possible mechanisms of the hESCs to be the molecules promoting repair of structures or return of function, including growth factors or neural transmitters, and the mechanical glue and bridging effect instead of direct cell replacement. Using the same model in 2010, the group evaluated a neural stem cell line (F3) and human bone marrow stem cells (B10) [142]. They found that bone marrow stem cells enhance the reclosure of NTDs. In contrast, the neural stem cell line (F3) lacked reclosure capabilities due to the poor survival rate of the stem cells.

Prenatal neural stem cells (NSCs) for the fetal repair of SB were introduced by Fauza et al. [143]. NSCs exert a protective effect with their neurotrophic and neuroprotective factors, thus promoting survival and regeneration of neural elements. Using the fetal lamb model, they found that NSCs retain an undifferentiated state and produce neurotrophic factors after delivery to the fetal spinal cord.

Human-induced pluripotent stem cells (iPSCs) from skin fibroblasts were introduced into the injured spinal cord in the fetal lamb model of MMC by Saadai et al. [144]. They found that the human-induced pluripotent-stem-cell-derived neural crest stem cells had greater than 95% viability and demonstrated neuronal differentiation in vitro. The iPSC neural crest stem cells survived, integrated, and differentiated into neuronal lineage in the fetal lamb model of MMC. Human-induced pluripotent stem cells, derived from amniotic fluid, were also applied to artificial skin in a rat model of MMC and showed differentiation into keratinocytes by adding Y-27632 and epidermal growth factor (EGF) [145].

Mesenchymal stromal cells (MSCs) from a variety of sources have more recently shown promise. Amniotic-fluid-derived MSC therapy was introduced as a potential prenatal therapy for a rat model of MMC by Abe et al. [146]. The proposed mechanism is the hepatocyte growth factor secretion leading to spinal cord coverage. In a rabbit model, Shieh et al. showed that a concentrated intra-amniotic injection of MSCs or trans-amniotic stem cell therapy (TRASCET) might be a promising option for prenatal management of SB [147]. Their studies show partial or complete skin-like coverage of SB in rodents and rabbits compared to the control groups.

Bone-marrow-derived mesenchymal stem cells (BM-MSCs) have also led to better reclosure of the spinal cord in rat models of SB, thought to be associated with brain-derived neurotrophic factor (BDNF) and neural growth factor (NGF) expression [148]. Subsequent studies by Li et al. showed increased epidermal growth factor (EGF) and fibroblast growth factor (FGF) in the transplanted cells [149].

Finally, our group has shown that in utero treatment with placenta-derived mesenchymal stromal cells (PMSC) improved motor function and increased the preservation of large neurons within the spinal cord in fetal ovine MMC models [150,151].

Most recently, a first in human FDA-approved Phase 1/2a clinical trial is applying clinical grade PMSCs, seeded on an FDA-approved extracellular matrix, to augment fetal surgery and repair of MMC. The Cellular Therapy for In Utero Repair of Myelomeningocele (CuRe) Trial is the first clinical trial utilizing stem cells for the fetal repair of MMC. The regenerative capacity of the fetal environment and PMSCs may provide the next leap in improving the outcomes for children afflicted with this devastating disease.

## 8. Conclusions

The most common survivable congenital defect of the central nervous system, SB, continues to push us to understand the cellular mechanisms of embryological development. While there have been major strides in understanding the steps in neurulation, we are still lacking in our ability to correct the deficits that lead to NTDs. Our current hope is for novel therapies and optimized medical and surgical management to provide improved outcomes and quality of life for these patients, until the complete biological underpinnings of embryonic development are so well-understood that prevention—and not intervention—would be the treatment of choice.

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
