# Peer review of "Spina Bifida: A Review of the Genetics, Pathophysiology and Emerging Cellular Therapies"

_jdb, 2022, doi:10.3390/jdb10020022_

Round 1

Reviewer 1 Report

The authors reviewed the knowledge on the pathogenesis on Myelomeningocele (MMC) and the possible new therapeutic treatments.

I suggest that the authors specify that MMC is only the most severe form, while many other different forms, mostly closed forms of NTDs, are nowadays more commonly present in the clinic.

In the introduction line 26 substitute "life-long paralysis" with motor impairment

line 28 "downward displacement of the posterior aspect..." should be changed with: displacement of the cerebellar vermis and tonsils. Furthermore the author should take in account that hydrocephalus and Chiari II malformation are 98% associated with MMC, please describe it better also at page 7.

Page 2 line 60 correct VANGL1 and VANGL2 all over.

Page 4: secondary neurulation is due to a mechanism of "retrogressive differentiation" please describe it better.

Page 6 line 273: "hind-brain" should be "hindbrain".

Page 7 line 292 correct "sever" in severe.

Reviewer 2 Report

The work promises a summary of the state of science on spina bifida.

Nevertheless, the sentence "remains unknown" (or similar) appears too often, this does not contribute to more transparency. If this statement was formulated in a certain context, the exact scientific question should be formulated. 

I would put the chapter embryology in the first place.

In detail:

line 30: delete "in extension"

line 32- 36 should be shortened

chapter 4: neurulation has been well described.  The consequences of disturbed primary and secondary neurulation should follow immediately after the description of normal development. Otherwise, it's just a list.

The chapter 2 " genetics has two subchapter: 3.1 and 3.2. Please correct.

I would prefer "spina bifida: a review" as the title.

The chapter 5, folate mechanismn should be rewritten more clearly. Maybe a table is suitable (study, model, findings, questions, year etc) with a short summary at the end.

line 204 -210 is this a summary? The sentences do not seem to say more than "we dont know enough" and "Spina bifida is a multifactorial event". Please rewrite.

Reviewer 3 Report

This review covers an important topic. However, there are several weaknesses that detract from its value to research or clinical specialists in spina bifida, as well as to the general reader interested in this congenital malformation. Despite the title of the review, it does not really address “The Cellular Level”. It does mention some genes and folate pathways that work at the cellular level, but only narrowly focuses on some genes in animals and humans, and does not describe how they work. There are several concerns with organization and editing, including manuscript citations in the References section. Several references are out-of-date and may not be accurate currently. The strongest sections of the review relate to prenatal surgery and research on prenatal stem cell repair of spina bifida, but these topics deviate from the title of the review.

Concerns:

  1. The manuscript should be more carefully reviewed. For example, in the first paragraph of the Introduction alone, some of the statements are not clear, accurate, contradict themselves, or may not be properly referenced:
  2. Line 23, sentence starting with “Most recent studied…” is not clear. Are the authors referring to the incidence of MMC or all NTDs (including anencephaly) as indicated in the reference title? The reference was published 2008, so data could be 20 years old, so may not be “recent”.
  3. Lines 23-25: This sentence contradicts the second sentence. One or the other should be revised.
  4. Line 26: The authors should add "Many of" to sentence starting "those who survive are afflicted..."
  5. Lines 27-29: The authors should add “which is associated with MMC”. Arnold-Chiari II malformations are not, strictly speaking, a closure defect, but appears to be secondary to a closure defect.
  6. Ref 1 is incorrect (see below).
  7. Line 32, sentence starting with “Unfortunately, the cause of SB remains largely unknown,” is not accurate. It reflects a weakness of the authors on the understanding of the pathogenesis of SB. First of all, there is not a single cause, secondly, there is much known (albeit, incompletely) about the mechanisms for various genetic, nutrient, and environmental causes of SB. The remainder of the sentence indicates that much has been revealed in recent decades. However, the second sentence of this paragraph (lines 34-36) references a paper published in 1973, more than 10 years before many genes associated with neural tube defects in mice were identified, and about 20 years before folic acid was identified to prevent many neural tube defects!. These kinds of statements and references cited raise concerns about the authors’ knowledge about current understanding of causes of SB.
  8. Based on the title, the review should include nongenetic, non-nutrient causes of, such as anticonvulsants, and maternal metabolism, such as diabetes and obesity.
  9. Section 3.1 (should be 2.1): Only discusses mouse genes involved in PCP. Many more genes & pathways have been identified (since 2014, the last reference cited in this section). This should be one of the strongest sections of the review, but it could benefit from development.
  10. Section 3.2 Human genetics (should be 2.2): Many of these references are 20+ years old. Have any of these genes been studied for mechanisms? In animal models? This should be one of the strongest sections of the review, but it could benefit from development.
  11. Section 4. Embryology (should be Section 3) should specify references that refer to mouse or human, because there are some differences at initiation and secondary neurulation. Also, sentences lines 143-146 need reference(s).
  12. Section 5. Folate Metabolism (should be Section 4):
  13. First paragraph: What cell? Intestinal? Where is 5-MTHF produced?
  14. Second paragraph: Transport of single carbons: From where to where? Or, are they converted into different 1C metabolites?
  15. Lines 194-196 is a 2004 reference to anti-folate receptor antibodies. How common are they?
  16. Section 6. Alternative Micronutrients (should be Section 5). This section only addresses Homocysteine, which is not a micronutrient. It should be included with the section on Folate Metabolism.
  17. Section 8.2 Bowel, bladder, and sexual dysfunction (should be Section 7.2). Does not address sexual dysfunction.
  18. Lines 348-349 “the now standard of care, fetal repair for eligible patients” is an overstatement. There are still many primary centers that do not have the infrastructure or skill levesl for fetal repair, and there are still concerns whether primary and secondary outcomes are beneficial relative to postnatal surgery. This is partially addressed in the following sentence (lines 349-352).
  19. Several errors in formatting in the References section including:
  20. Ref 1 published in BDRA. "A" is missing after "Part"
  21. Some citations only include authors’ initials, not surnames.
  22. Some citations only include authors’ given names, and initials for surnames.
  23. There may be other errors in referencing formatting, but these were the ones that I noted. There is reference management software that can prevent these errors.

Round 2

Reviewer 3 Report

The authors have revised the manuscript, including deleting part of the title to be more consistent with content, reorganization of some sections, and attempting to make the Introduction more current. However, there continue to be several concerns with this manuscript. A review does not need to advance the status of knowledge in the field, but it should be authoritative, accurate, and current. There continue to be statements that are inaccurate or not relevant to SB, and incorrect or out of date reference citations. A reviewer should not need to fact check all of the references of a manuscript because of high rates of errors and mis-citation. Additionally, as was recommended, the authors stated that they have implemented reference management software. However, reference management software will not prevent errors in the References section, as can occur if the reference library is generated manually rather than by using the software import function, and saving as the appropriate reference type, or if the incorrect reference is inserted into the text. The authors also need to be sure to format the references according to the style used by the journal.

The concerns are listed mostly in order of appearance, rather than major and minor.

  1. Deletion of "at The Cellular Level" from the title is more consistent with the actual content of the review, but don't the authors want to elaborate on what it covers or contributes? For example, a PubMed search generated 999 reviews with "Spina Bifida" in the title or text. A more focused title may help readers find this one. (Also, please note that Keywords should not repeat words in the title, as words in the title are included by default as search terms.)
  2. Intrauterine repair of spina bifida is referred to in the Abstract, and twice in the manuscript, as the "standard of care". In the prior review, I felt that “standard of care” is an overstatement. I continue to object to this language, despite the addition of "Moreover, there are still many centers in the country that lack the infrastructure, training, and support stuff [sic] necessary; therefore, while fetal intervention offers hope, it is still a highly specialized intervention with limited global spread. ". In medical terms, a "standard of care" indicates that it is accepted by medical experts as a proper treatment that is widely used by practitioners in that discipline. This is not the case for intrauterine SB repair. There are about a dozen centers in the USA that can perform this surgery, but many more centers that perform post-natal repairs and are not planning to move to intrauterine surgery. Many neurosurgeons are not convinced that the outcomes of fetal repair are superior to conventional post-natal repair to outweigh the risks, and many clinical centers lack the infrastructure to develop the multidisciplinary physical resources. It should be noted that, "standard of care" has legal implications if it is not offered. The authors could down-play their statements by either deleting "standard of care", or inserting "what may become", or "state of the art", or something similar.
  3. There are References 4 and 5 in the References section, but, as far as I could tell, these references are not cited in the manuscript.
  4. Lines 27-28: Reference 6 (mortality rate) is from 1997 and is probably not true now.
  5. Lines 35-36, “Our understanding of NTDs has improved…” Something is missing. The etiology of NTDs? The causes? The mechanisms for formation?
  6. Line 39: “congenital disorder (or malformation)” is preferable to “congenital disease”.
  7. Lines 39-41, sentence starting with, ”It appears that..." is not true. There can be single perturbations, and not necessarily cumulative. Rather, there can be multiple causes, as stated in the previous sentence. This sentence could be deleted. If it is kept, Ref 8 for this statement refers to mutant mouse models and is not correct for this sentence.
  8. Lines 42-43, “the genetic locus in humans remains for future study.” “Locus” should be “loci”, because there are already several loci identified. This area does not remain “for future study”, as there are several risk genes already published within the past 20 years. Rather, it would be correct to state that identification of genetic loci is an ongoing area of study. Ref 8 for this statement refers to mutant mouse models and is not correct for this sentence.
  9. Lines 43-51 cites Ref 10, which is not the correct reference. Ref 8 belongs to this sentence. However, Ref 8, which was published in 2007, referred to "in excess of 190", not "approximately 250". Ref 23, from the same authors as Ref 8 (in 2010) is cited later in the paper and says, "now exceeds 240". It is recommended to delete Ref 8 and use Ref 23. The authors should indicate whether these 240+ mutations all spontaneous or induced (by chemical- or radiation-induced mutagenesis), or if they include genetically engineered (by transgenic or knockout technology) mutations.
  10. Ref 10 appears to be a book chapter and should be properly formatted in the References section.
  11. Line 63: “can result in” is more accurate than “results in”
  12. Lines 66-67: Ref 12 is an undergraduate (college) textbook. The authors should cite peer-reviewed citations of these processes, which might include review articles. (Such references might be cited in the textbook.)
  13. Lines 68-78: This paragraph should be reorganized, including to clarify that it describes the mouse. There are differences in human embryos, which should be described.
  14. The sentences preceding Ref 13 describe primary neurulation, but Ref 13 describes secondary neurulation.
  15. Lines 82-86 are not correct. Exencephaly is a precursor, less severe form, of anencephaly, not caused by different closure sites.
  16. Lines 89-95, secondary neurulation: Please state if this describes human or mouse secondary neurulation. Both a human and a mouse reference are cited. Lines 96-99 refers to the day when this process occurs in human embryos, but the reference is to mouse embryos.
  17. Lines 91-92, “Canalization is the process by which the neural tube elongates toward the posterior neuropore.” From which direction?
  18. Lines 116-118 cite Ref 22 from 2004 and may not be correct now due to more recent publications.
  19. Lines 120-123: It is advised to remove "similar to those in humans" since it is not clear what is meant.

It is advised to change "a few" to "some" (more than a few have been identified); "biochemical pathways" to "cellular processes" (PCP and apoptosis are not biochemical pathways); "are involved" to "have been identified".

  1. Line 124: What is meant by "Polarization"? Apical-basal? Internal-external? Electrical? Which mature tissues?
  2. Refs 24 and 25 (which are more than 20 years old) are not correct citations for these genes.
  3. Line 137: Ref 27 is a human study, not mouse, and this section is about NTDs in animal models.
  4. Paragraph lines 155-160 is not relevant. The MFH-1 reference cited is not involved in NTDs. The Traf4 gene is involved in formation of several structures, not specific to NTDs. From what I could see, Ref 34 doesn't deal with Traf4 mutation.
  5. Lines 161-162, “Lastly, genetic mutations in the canonical Wnt/ beta-catenin pathway have been studied in SB mouse model and shown to be implicated in human SB.” Wnt signaling IS involved in the PCP pathway, which was described earlier in this section.
  6. Lines 162-165, referring to LRP6, referenced by Ref 35 are not relevant, as they deal with osteo- and skeletal defects, not NTDs.
  7. All of the literature cited in the section on animal models deal with a single process (PCP), or genes that are unrelated to NTD or have more widespread effects than just NTDs. The authors deleted text on Pax3 (related to folate metabolism), but it encodes a transcription factor and PAX3 also has antiapoptotic effects in the neural tube. In the section on Human Genetics, the authors indicate that whole exome sequencing of human subjects identified Shroom3 (involved in PCP), Pax3, Grhl3 (causes a proliferation defect). These genes are all associated with NTDs in mice and should be discussed in this section.
  8. Lines 176-179: This sentence seems out of place here. It is advised to place after the descriptions of single genes and elaborate.
  9. Lines 201-203: The CCL2 gene is associated with NTDs, but Ref 46 doesn't show this.
  10. Lines 203-211 and Ref 47: CCL2 gene variants are associated with SB independent of fever.
  11. Line 213: Advised to change "attributed this to" “speculated that this was due to”.
  12. Ref 48 is incomplete in the References section.
  13. Refs. 49 & 50 did not study gene mutations of enzymes involved in homocysteine-folate metabolism.
  14. Ref 52, the surname of the first author (who is a co-author in some of the other cited manuscripts) is "De Marco", not "Marco".
  15. Lines 249-254: Refs 66-68 deal with gestational diabetes and obesity (which may be associated with type 2 diabetes). Historically, type 1 diabetes has long been known to increase risk for NTDs. The authors should include type 1 diabetes reference(s).
  16. Lines 254-256: Refs 69 & 70 only studied vitamin B12, not folate, zinc, or vitamin C.
  17. Lines 256-257: It is not accurate to call caffeine a "notable" risk factor. From the Abstract of Ref 71: "CONCLUSIONS: Additional studies should confirm whether women who consume caffeine are at increased risk for pregnancies complicated by NTDs."
  18. Lines 257-258: Earlier in the manuscript, the authors used Ref 47 (from 1998) as evidence that hyperthermia increased risk for SB. The authors should use that reference, instead of Ref 73 (1978), which did not report an effect of hyperthermia on NTDs, and start the sentence with "As noted above,".
  19. Lines 267-269: Valproic acid has been shown to have activity as a histone deacetylase inhibitor, but Ref 76 does not show that this leads to neurulation failure as indicated in the text. Ref 77 does not study valproic acid or histone deacetylase activity.
  20. Lines 273-276: It is advised to delete the second part of the sentence and Refs 80 & 81 related to effects of folic acid on cognitive function of AED-exposed offspring. These children did not have NTDs, and the pathogenesis and timing of exposure related to cognitive development may be unrelated to NTDS.
  21. Lines 332-341: Ref 82 is a book chapter and should be formatted appropriately. The pathway described is not accurate. Folic acid must be converted to dihydrofolate, which is converted to tetrahydrofolate, which takes place not just in hepatocytes.
  22. Ref 84 is incomplete.
  23. Line 378: Neural crest is not endoderm.
  24. Lines 376-380: Irrelevant. None of these structures exist at the time of neural tube formation.
  25. Line 414: Methylation of what (regarding epigenetic processes)?
  26. The first author of Ref 99 is "van der Linden", not "Linden". "van" is part of some of the other authors' surnames in that reference and in some others.
  27. Lines 445-454: Ref 105 refers to mutations in mothers, not patients. Ref 106 (from 1976) does not support this statement (and, in fact, is a commentary about a researcher in the 1960's and 1970's who hypothesized that undernutrition could be a cause of NTD). Why are Refs 105 & 106 cited before Refs 103 and 104?
  28. Lines 461-462: “Today, the decreased incidence of NTDs is largely due to the prevention of folate deficiency, a risk factor that was not fully appreciated until the turn of the 20th century.” This is not true. First of all, the turn of the 20th century was in 1900. I think the authors meant the turn of the 21st century, but this is also not true. This was recognized in the late 1980’s-early 1990’s.
  29. Lines 463-466: Numerous errors in references for this statement. Refs 108 (1981) used multivitamins and precedes studies specifically on folic acid; Ref 110 used multivitamins.
  30. Lines 468-469: Ref 112 is from 1996. Ref 113 is from the CDC (not the FDA) and is from 1992. There is no reference for a mandate.
  31. Lines 472-477. This is a difficult sentence to understand. I think the authors mean that there are some folate-independent NTDs. This should be more clearly stated.
  32. Lines 481-485. Please specify that the studies reported in Refs 115 & 116 were performed using fetal rats and pigs.
  33. Lines 486-489: “The work by Meuli et al. in mutant mouse models has supported this hypothesis, in addition to clinical observations on early gestation human fetuses, which demonstrates progressive loss of fetal leg movement in utero[117].” Meuli is not the first author of Ref 117 (although he is an author), so “Meuli, et al.” is incorrect. This paper did not study human fetuses, so “in addition to clinical observations on early gestation human fetuses, which demonstrates progressive loss of fetal leg movement in utero” lacks a reference.
  34. Lines 492-495. The authors should specify that the study in Ref 118 was performed using fetal lambs.
  35. Lines 497-498: “There is no medical treatment for the constellation of symptoms that develop associated with SB and MMC.” I don't understand what is meant by this. Isn't surgery a medical treatment? Furthermore, Lines 522-525 do refer to medical treatments.
  36. Line 499: “Initial management involves basic resuscitation…” Are babies born unconscious?
  37. Lines 506-512: As written, it appears that hydrocephaly develops only with MMC + Chiari II malformation, when, in actuality, it often occurs with SB that is not characterized as MMC + Chiari II malformation.
  38. Line 521: What is meant by “level of disruption”? Severity? Spinal cord level? If the former, I suggest replacing “level” with “severity”. If the latter, I suggest changing to “Depending on the spinal cord level(s) that is disrupted, many…”
  39. Lines 547-548: Ref 125 doesn't deal with stance and ambulation, but upper limb function.
  40. Lines 548-550: Ref 126 is a 2001 review on fetal surgery, on which the senior author of this manuscript is one of the authors. It is not the original source for the statement in lines 548-550 but may contain references to the original sources.
  41. Lines 552-554: Refs 127-129 are from 2001, 1990, and 1995 and may not be current.
  42. Lines 565-566: “For the past century, the initial management of SB has been surgical closure within the first 48 hours of birth.” Needs a reference.
  43. Line 610: “NTDs” should be “NTD” (singular).
  44. Lines 617 and 621: Please provide the specie(s) from which these stem cell lines were derived.
  45. Lines 626-629: The in vivo fate of the hiPSC-derived neural crest stem cells after engraphment into the injured spinal cord in the fetal lamb model of MMC should be indicated.
  46. Lines 629-631: The authors should specify that this is a rat model of MMC (they refer to a rat model, but not MCC) and the iPSC were amniotic fluid-derived.
  47. Lines 632-647: For all of the statements that use Refs 142-147, the species sources of the MSC and the species with MMC should be indicated.
  48. Line 635: "factor" is missing.
  49. The Reference entry for Ref 147 is not correct. It was published in 2017, not 2018.
  50. Line 659: "neural defects" should be changed to "NTDs".
  51. Lines 672-672 (Acknowledgements): The sentence lacks a subject.
